# Effects of Different Therapeutic Approaches on Redox Balance in Psoriatic Patients

**DOI:** 10.3390/biomedicines12030587

**Published:** 2024-03-06

**Authors:** Marija V. Medovic, Vesna M. Milicic, Ana B. Ravic Nikolic, Gordana J. Ristic, Rasa H. Medovic, Marina R. Nikolic, Aleksandra Z. Stojanovic, Sergey B. Bolevich, Natalia G. Bondarchuk, Alexander A. Gorbunov, Slobodanka L. Mitrovic, Vladimir Lj. Jakovljevic, Ivan M. Srejovic

**Affiliations:** 1Department of Dermatovenerology, Faculty of Medical Sciences, University of Kragujevac, Svetozara Markovica 69, 34000 Kragujevac, Serbia; makastojanovic88@gmail.com (M.V.M.); vesna.milicic.kg@gmail.com (V.M.M.); anaravic74@gmail.com (A.B.R.N.); gordana.ristic1503@gmail.com (G.J.R.); 2University Clinical Center Kragujevac, Zmaj Jovina 30, 34000 Kragujevac, Serbia; rasamedovic@gmail.com; 3Department of Pediatrics, Faculty of Medical Sciences, University of Kragujevac, Svetozara Markovica 69, 34000 Kragujevac, Serbia; 4Department of Physiology, Faculty of Medical Sciences, University of Kragujevac, Svetozara Markovica 69, 34000 Kragujevac, Serbia; marina.rankovic.95@gmail.com (M.R.N.); drvladakgbg@yahoo.com (V.L.J.); 5Center of Excellence for the Study of Redox Balance in Cardiovascular and Metabolic Disorders, University of Kragujevac, Svetozara Markovica 69, 34000 Kragujevac, Serbia; vranicaleksandra90@gmail.com; 6Department of Pharmacy, Faculty of Medical Sciences, University of Kragujevac, Svetozara Markovica 69, 34000 Kragujevac, Serbia; 7Department of Human Pathology, First Moscow State Medical University I.M. Sechenov, Trubetskaya Street 8, Str. 2, 119991 Moscow, Russia; bolevich2011@yandex.ru; 8Department of Pharmacology, First Moscow State Medical University I.M. Sechenov, Trubetskaya Street 8, Str. 2, 119991 Moscow, Russia; bondarchuk_n_g@staff.sechenov.ru (N.G.B.); gorbunov_a_a@staff.sechenov.ru (A.A.G.); 9Department of Pathology, Faculty of Medical Sciences, University of Kragujevac, Svetozara Markovica 69, 34000 Kragujevac, Serbia; smitrovic@fmn.kg.ac.rs

**Keywords:** psoriasis, oxidative stress, biologics, secukinumab, ustekinumab, methotrexate

## Abstract

Given that oxidative stress represents an important etiological factor in the pathogenesis of psoriasis, the aim of this study was to assess the effects of different therapeutic approaches, methotrexate, secukinumab, and ustekinumab on systemic oxidative stress biomarkers in psoriatic patients. This study involved 78 psoriatic patients, divided into the group treated with methotrexate (23 patients), secukinumab (28 patients), and ustekinumab (27 patients), and 15 healthy controls. Oxidative stress biomarkers (index of lipid peroxidation measured as TBARS, nitrites (NO_2_^−^), superoxide anion radical (O_2_^−^), and hydrogen peroxide (H_2_O_2_)) and antioxidative defense system (superoxide dismutase (SOD) activity, catalase (CAT) activity, and reduced glutathione (GSH)) were determined spectrophotometrically from the blood before the initiation of therapy in 16th, 28th, and 52nd week. O_2_^−^ and SOD showed the most prominent changes comparing the psoriatic patients and healthy controls. CAT activity was significantly lower in psoriatic patients, and methotrexate induced a further decline in CAT activity. Ustekinumab induced a significant increase in GSH level after 52 weeks of treatment, while methotrexate reduced GSH. All applied therapeutic options induced a reduction in PASI, BSA, DLQI, and EARP. Biological drugs exert more pronounced antioxidant effects compared to methotrexate, which is most clearly observed in the values of O_2_^−^ and SOD.

## 1. Introduction

Psoriasis is defined as a chronic, recurrent, inflammatory, immune-mediated noncontagious disease characterized by skin lesions usually accompanied by systemic complications. The most common form of psoriasis is psoriasis vulgaris, which causes well-defined, symmetrical, sharply demarcated erythematous skin lesions with overlying hyperkeratotic plaques that can be found in different areas, but they most often occur in predilection locations, such as the knees, elbows, scalp, and trunk [1,2]. Skin lesions are usually accompanied by itching, pain, and cracking, with consequent bleeding and flaking of the skin. Due to the autoimmune and inflammatory etiology, the usual clinical course of psoriasis is often complicated by disorders of many organ systems, including psoriatic arthritis, cardiometabolic diseases, gastrointestinal and urogenital disorders, infections, and malignancies [3,4]. It is assumed that various external stimuli, such as stress, trauma, or infection, initiate immune-mediated aberrant differentiation of keratinocytes and disruption of skin homeostasis [5,6]. Although previously defined as a Th1-mediated disease, now it is shown that Th17 and Th22 cells are also crucial in the development of psoriatic lesions [7,8]. Increased production of pro-inflammatory cytokines by T cells resulted in increased levels of interferon-γ (IFN-γ), interleukin-17 (IL-17), and IL-22 in psoriatic skin changes [9]. IL-23 appears to be one of the most important mediators in the development of psoriasis due to its ability to shift naïve T cells to pro-inflammatory Th17 cells and stimulate the production of IL-17 and IL-22 [10,11]. A deeper understanding of immune disruptions in psoriasis enabled the development of novel biological drugs, which target specific cytokine pathways involved in psoriasis development and thus achieve significantly better outcomes. Biological drugs target various cytokine pathways related to the pathogenesis of psoriasis, such as tumor necrosis factor α (TNF-α) (adalimumab, certolizumab, etanercept, infliximab), the p40 subunit of IL-12 and IL-23 (ustekinumab), IL-17A (ixekizumab, secukinumab), IL-17 receptor (brodalumab), and the p19 subunit of IL-23 (guselkumab, risankizumab, and tildrakizumab) [12]. Interestingly, although it has been shown that IL-1, IL-6, and IFN-γ are important in the development of psoriasis, their inhibition did not achieve a significant clinical effect [13]. Common therapeutic options before the introduction of biological drugs were methotrexate (MTX), cyclosporine, dexamethasone, or salicylic acid. MTX is therapeutic option number one for many physicians due to its efficacy and affordability. The mechanism of action of MTX is based on the inhibition of dihydrofolate reductase and JAK/STAT (Janus kinase/Signal Transducer and Activator of Transcription proteins) pathway activity, resulting in the reduction in immune response and keratinocyte proliferation [14].

Oxidative stress is usually defined as a disproportion in the production of reactive species (mostly oxygen and nitrogen) and their elimination and neutralization by the antioxidative defense system. Various cellular structures and molecules are vulnerable to oxidative damage and because of which, oxidative stress is an important link in the development of many diseases. Oxidative stress is recognized as a step in the pathophysiological cascade of psoriasis [6,15,16]. An induction of oxidative stress leads to an increased production of IL-23A in human Langerhans-like cells, which is proposed as a mechanism of immunoregulation of cutaneous dendritic cells and an induction of inflammation [17]. Furthermore, IL-17A appears to be one of the most prominent cytokines involved in the pathogenesis of psoriasis. IL-17A increases the production of IL-26 and IL-29 by Th17 cells and affects the release of IL-19, IL-24, IL-26, and IL-36 [6,18,19,20]. Such a pro-inflammatory milieu induces an increased production of reactive oxygen species (ROS) and oxidative damage. ROS may act as chemoattractants, resulting in neutrophil mobilization and accumulation in the skin [21]. The activation of neutrophils results in further augmentation of oxidative damage and inflammation. 

Given the close interconnection between inflammation and oxidative stress in psoriasis, the aim of this study was to assess the effects of different therapeutic approaches in the treatment of psoriasis on redox balance. We compared the values of prooxidants and elements of the antioxidative defense system in patients treated with different biologics and MTX.

## 2. Materials and Methods

### 2.1. Patients 

A total of 78 patients with diagnosed psoriasis were included in this study. Patients were admitted to the Department of Dermatology, University Clinical Center Kragujevac (Kragujevac, Serbia). This study was conducted as a prospective, clinical study, lasting 18 months. 

All patients were divided into three different subgroups according to the established therapeutic protocol.

Group 1: A total of 23 patients (mean age 46.70 ± 14.65 years; 16 men and 7 women) with moderate or severe plaque psoriasis who required oral immunosuppressive therapy. The dose of MTX did not exceed 20 mg per week. MTX-treated patients received the drug once a week in the recommended dose.

Group 2: A total of 28 patients (mean age 46.04 ± 13.95 years; 20 men and 8 women) in whom previous therapeutic options had not been clinically effective or had adverse reactions to previously applied therapy who met the conditions for the use of secukinumab (300 mg s.c. per therapeutic dose). Secukinumab-treated patients received the first 5 doses in 5 weeks (one dose per week); afterwards, the maintenance phase involved the application of the drug once every 4 weeks.

Group 3: A total of 27 patients (mean age 46.00 ± 16.20 years; 18 men and 9 women) in whom previous therapeutic options had not been clinically effective or had adverse reactions to previously applied therapy who met the conditions for the use of ustekinumab in a dose of 45 mg s.c. or 90 mg s.c., depending on the weight of the patients (patients weighing up to 100 kg received a dose of 45 mg, patients weighing over 100 kg received a dose of 90 mg). Ustekinumab-treated patients received the first 2 doses 4 weeks apart; afterwards, the maintenance phase involved the application of the drug once every 12 weeks.

The inclusion criterion for all patients was diagnosed psoriasis. A diagnosis of psoriasis was established according to the Psoriasis Area Severity Index (PASI) and Body Surface Area (BSA) [22]. Skin biopsy and clinical examination were conducted in the case of an unclear clinical picture to exclude other skin diseases. 

The inclusion criterion for MTX administration was patients with moderate to severe psoriasis (determined according to the PASI > 10 and BSA > 10) who were previously unsuccessfully treated with topical drugs.

For patients treated with secukinumab and ustekinumab, the inclusion criteria were a diagnosis of a moderate or severe form of psoriasis (determined according to the PASI > 10 and BSA > 10) and previous use of other forms of systemic therapy (mainly MTX) to which they had a poor therapeutic response or a type of adverse reaction. The interval between the use of other forms of therapy and biological drugs (secukinumab and ustekinumab) was at least 10 weeks. Patients were randomly assigned to groups treated with secukinumab or ustekinumab.

The exclusion criteria were pregnancy and/or nursing, age below 18 or above 75 years, the presence of active or latent tuberculosis, malignancy, hepatitis B and/or C, HIV infection, renal or liver insufficiency, cardiovascular diseases (myocardial infarction, implanted stent, bypass, pacemaker or artificial heart valve, unstable angina pectoris), and unregulated diabetes or diabetes accompanied with severe macro- or microangiopathic complications.

### 2.2. Healthy Controls

A total of 15 age-matched subjects (mean age 44.27 ± 11.61 years; 10 men and 5 women) were included in this study as healthy controls. They were admitted to the Department of Dermatology, University Clinical Center Kragujevac (Kragujevac, Serbia) due to the application of liquid nitrogen for the treatment of verrucae vulgares or seborrheic keratosis.

The exclusion criteria for healthy controls were any dermatological disease except verrucae vulgares or seborrheic keratosis, pregnancy and/or nursing, age below 17 or above 75 years, the presence of active or latent tuberculosis, malignancy, hepatitis B and/or C, HIV infection, renal or liver insufficiency, cardiovascular diseases (myocardial infarction, implanted stent, bypass, pacemaker or artificial heart valve, unstable angina pectoris), and unregulated diabetes or diabetes accompanied with severe macro or microangiopathic complications.

Patients in the psoriasis groups, as well as healthy controls, were free of any medication known to affect redox status, and a detailed medical history, including all current medications, the presence of other diseases, smoking, and level of education, was recorded.

Written informed consent was obtained from all participants, and the study protocol was approved by the Ethics Committee of the University Clinical Center Kragujevac prior to the onset of this study (approval number 01/20-808, 4 November 2020). The investigation was conducted in accordance with the principles outlined in the Declaration of Helsinki and the principles of Good Clinical Practice (GCP).

### 2.3. Disease Activity Outcomes 

The *Psoriasis Area Severity Index* (PASI) is an index used to express the severity of psoriasis. It combines severity (erythema, induration, and desquamation) and the percentage of the affected area. Clinical signs are marked on a scale from 0 (none), 1 (slight), 2 (moderate), 3 (severe), to 4 (very severe). To obtain a PASI score, for each part of the body, the points assigned for the three severity signs are summed and then multiplied by the involvement score and the severity of that part (i.e., 0.1 for the head, 0.2 for hands, 0.3 for trunk, and 0.4 for legs) [23]. 

*Body Surface Area* (BSA) is the arithmetic mean of the affected skin surface based on the assumption that the head (H) represents 10%, the upper limb (U) 20%, the trunk (T) 30%, and the lower limb (L) 40% of the total Body Surface Area. The formula for calculating BSA is as follows:

BSA = 0.1 × BSAH + 0.2 × BSAU + 0.3 × BSAT + 0.4 × BSAL. The severity of the disease was determined using mild psoriasis BSA ≤ 10 and moderate and severe psoriasis BSA > 10 [24].

### 2.4. Patient Reported Outcomes

The *Dermatology Life Quality Index* (DLQI) is a questionnaire consisting of ten questions and is used to measure the impact of skin disease on the quality of life. It is designed for people over 16 years of age. Questions cover the following topics: symptoms, shame, shopping and home care, clothing, social activities and leisure, sports, work or study, close relationships, sex, and treatment. The questions refer to the impact of the skin disease on the patient’s life during the previous week. Interpretation of the results is as follows: 0–1 = no impact on the patient’s life, 2–5 = little impact on the patient’s life, 6–10 = moderate impact on the patient’s life, 11–20 = very high impact on the patient’s life, 21–30 = extremely great impact on the patient’s life [25].

*Early Arthritis for Psoriatic Patients* (EARP) is a questionnaire consisting of 10 questions and was developed to facilitate the diagnosis of psoriatic arthritis (PsA) and its differentiation from other forms of arthritis [26]. During patient follow-up, a questionnaire was used to identify all patients with joint pain, as well as monitor the effect of therapy on the joints (pain, stiffness, reduced mobility).

### 2.5. Blood Sampling

Blood sampling was obtained in the same manner for all participants in this study at the Department of Dermatology, University Clinical Center Kragujevac (Kragujevac, Serbia). Venous blood from all participants was collected between 8 and 10 a.m. following at least a 10 h fasting period in a quiet, air-conditioned, and temperature-controlled room (22–24 °C). Blood was collected in Vacutainer tubes containing 0.129M sodium citrate (BD Vacutainer Blood Collection System). Blood was centrifuged to separate plasma and red blood cells (RBCs) and stored at −80 °C.

Blood sampling was performed before the beginning of therapy for all psoriasis-suffering patients and healthy controls (week 0), and then in the 16th, 28th, and 52nd week after beginning therapy.

### 2.6. Redox Status 

Redox status was measured spectrophotometrically by measuring the index of lipid peroxidation measured as Thiobarbituric Acid Reactive Substances (TBARS), nitrites (NO_2_^−^), superoxide anion radical (O_2_^−^), and hydrogen peroxide (H_2_O_2_) in plasma (as previously described in [27,28,29]). Activities of the corresponding antioxidative enzymes superoxide dismutase (SOD), catalase (CAT), and the concentration of reduced glutathione (GSH) were measured spectrophotometrically in erythrocyte lysates (as previously described in [27,28,29]).

### 2.7. Determination of Index of Lipid Peroxidation measured as Thiobarbituric Acid Reactive Substances (TBARS)

The degree of lipid peroxidation in plasma was assessed by TBARS measured using 0.4 mL 1% thiobarbituric acid (TBA) in 0.05 NaOH mixed with 0.8 mL of plasma, incubated at 100 °C for 15 min, and measured at 530 nm. Distilled water was used as a blank probe. TBA extract was obtained by combining 0.8 mL plasma and 0.4 mL trichloroacetic acid. Thereafter, the samples were put on ice for 10 min and centrifuged for 15 min at 6000 rpm. Distilled water was used as a blank probe.

### 2.8. Nitrite Determination 

NO decomposes rapidly to form stable metabolite nitrite/nitrate products. The method for the detection of plasma NO_2_^−^ levels is based on the Griess reaction. NO_2_^−^ was determined as an index of NO production with Griess reagent (form purple diazocomplex). A total of 0.1 mL 3N perchloric acid, 0.4 mL 20 mM ethylenediaminetetraacetic acid, and 0.2 mL plasma were put on ice for 15 min and then centrifuged for 15 min at 6000 rpm. After pouring off the supernatant, 220 μL K_2_CO_3_ was added. NO_2_^−^ was measured at 550 nm. Distilled water was used as a blank probe.

### 2.9. Superoxide Anion Radical Determination

The level of O_2_^−^ was measured using a Nitro Blue Tetrazolium reaction in a TRIS buffer with plasma and read at 550 nm. Distilled water was used as a blank probe.

### 2.10. Hydrogen Peroxide Determination

The determination of H_2_O_2_ concentration is based on the oxidation of phenol red using H_2_O_2_ in the reaction catalyzed by the enzyme peroxidase from horseradish (POD). A total of 200 μL samples with 800 μL phenol red solution and 10 μL POD were combined (1:20) and measured at 610 nm.

### 2.11. Determination of Catalase, Superoxide Dismutase, and Reduced Glutathione

Isolated erythrocytes were washed three times with 3 volumes of ice-cold saline, and erythrocyte lysates were prepared according to McCord and Fridovich. Erythrocyte lysates were used for the determination of superoxide dismutase (SOD) and catalase (CAT) activity, as well as the amount of GSH. The determination of SOD activity is based on the epinephrine method of Misra and Fridovich. A 100 μL erythrocyte lysate and 1 mL carbonate buffer were mixed, and then epinephrine in a volume of 100 μL was added. Detection was performed at 470 nm. For the determination of CAT activity, erythrocyte lysates were diluted with distilled water (1:7) and treated with chloroform/ethanol (0.6:1) to remove hemoglobin. Then, 50 μL CAT buffer, 100 μL sample, and 1 mL 10 mM H_2_O_2_ were added to the samples. Detection was performed at 360 nm. The level of reduced glutathione (GSH) concentration was determined based on GSH oxidation with 5.5-dithiobis-6.2-nitrobenzoic acid using the Beutler method. The measurement of the absorbance is carried out at a wavelength of maximum absorption of 420 nm. Distilled water was used as a blank probe.

### 2.12. Statistical Analysis

All data were analyzed using GraphPad Prism 8 (Version for Windows, GraphPad Software, La Jolla, CA, USA). The results are expressed as means ± standard deviation of the mean (SD), median, or percentages, depending on the data type. The distribution of the data was checked by the Shapiro–Wilk test. An independent samples *t*-test (parametric) and a Mann–Whitney U test (nonparametric), as well as a one-way ANOVA and a Kruskal–Wallis test, were used to assess the difference in estimated variables between the groups. A *p*-value < 0.05 was regarded as statistically significant.

## 3. Results

This study involved 78 psoriasis-suffering patients divided into three groups according to the applied therapeutic procedure—MTX-treated patients, secukinumab-treated patients, and ustekinumab-treated patients. Patients in all groups were of similar age with similar gender distribution. In addition to psoriasis patients, 15 healthy patients of similar age and gender distribution were also included in this study. Table 1 shows the basic characteristics of the psoriatic patients and healthy controls.

All applied therapeutic procedures induced an improvement in the clinical presentation of the disease, as well as a resolution of histopathological marks of psoriasis. Figure 1 shows the clinical changes of psoriatic lesions since the diagnosis and evaluation of the patients, and then at the 16th, 28th, and 56th week from the beginning of therapy (MTX, secukinumab, and ustekinumab). Furthermore, Figure 1 shows the histopathological changes of psoriatic lesions before the introduction of therapy and after 52 weeks.

### 3.1. Disease Activity Outcomes

PASI and BSA scores were significantly reduced in all groups of patients after the 16th week of therapy, and values after the 16th week (28th and 52nd week) did not differ significantly compared to the 16th week (Figure 2(a1,a2) for MTX, Figure 2(b1,b2) for secukinumab, Figure 2(c1,c2) for ustekinumab).

### 3.2. Patient Reported Outcomes

Similar to the disease activity outcomes, EARP and DLQI significantly reduced after the 16th week of treatment in all groups of psoriatic patients, but they did not change significantly at subsequent assessments (28th and 52nd week) (Figure 2(a3,a4) for MTX, Figure 2(b3,b4) for secukinumab, Figure 2(c3,c4) for ustekinumab).

### 3.3. Values of Superoxide Anion Radical (O_2_^−^) and Superoxide Dismutase (SOD)

Levels of O_2_^−^ and SOD showed the most pronounced changes when comparing values between groups and within the same group (Figure 3). In all groups of patients with psoriasis, the values of O_2_^−^ were significantly higher compared to the healthy controls (Figure 3a). On the other hand, the activity of SOD was significantly lower in the groups of patients with psoriasis compared to the healthy controls (Figure 3b). All applied therapeutic procedures induced a significant reduction in O_2_^−^ levels, especially if the values before the introduction of therapy and after the 52nd week were compared (Figure 3a). Both applied biological drugs, secukinumab and ustekinumab, induced an increase in SOD activity, while MTX induced a further reduction in SOD activity (Figure 3b). O_2_^−^ and SOD values between groups were compared before the initiation of therapy and in the 52nd week. Comparisons between groups are shown in Table 2. Values of O_2_^−^ were significantly higher in psoriatic patients compared to healthy controls before the initiation of therapy, while at week 52, there were no differences between the groups. SOD activity was significantly lower in psoriatic patients in comparison to healthy individuals and MTX induced a further reduction in SOD activity, but even in the groups treated with biological drugs, SOD activity did not reach the values in the control group.

### 3.4. Values of Hydrogen Peroxide (H_2_O_2_) and Catalase (CAT)

Values of H_2_O_2_ were similar in all groups, healthy controls, and psoriatic patients, and the applied therapeutic procedures did not induce any significant change in the H_2_O_2_ level (Figure 4a). CAT activity was significantly higher in the control group in comparison to all groups of patients with psoriasis. Biological drugs did not induce a significant change in CAT activity, while MTX induced a reduction in CAT activity (Figure 4b). H_2_O_2_ and CAT values between groups were compared before the beginning of therapy and in the 52nd week. Comparisons between groups are shown in Table 3. H_2_O_2_ level was significantly higher in the psoriatic groups of patients in comparison to the control group, and these values remained higher even at the last measurement point. CAT activity was lower in patients with psoriasis, and therapeutic procedures did not significantly change it, except for MTX, which further reduced CAT activity. Ustekinumab induced a slight increase in CAT activity and as a result, in the 52nd week, catalase activity was significantly higher in this group compared to patients treated with MTX and secukinumab.

### 3.5. Values of TBARS, Nitrites (NO_2_^−^), and Reduced Glutathione (GSH)

The index of lipid peroxidation, measured as TBARS, was on a similar level in all groups. MTX and ustekinumab induced a significant reduction in TBARS after 52 weeks of treatment (Figure 5a). Values of NO_2_^−^, as a marker of nitric oxide (NO) production, did not differ significantly between groups and did not change significantly during the application of the examined therapeutic procedures (Figure 5b). GSH values were similar between all groups. MTX induced a reduction in GSH values after 52 weeks of treatment, while ustekinumab induced a significant improvement of GSH after 52 weeks of therapy (Figure 5c). TBARS, NO_2_^−^, and GSH values between groups were compared before the beginning of therapy and in the 52nd week. Comparisons between groups are shown in Table 4. The values of TBARS and NO_2_^−^ did not change significantly, and there were no differences between groups. GSH level in the MTX-treated group was higher compared to the other groups, but MTX induced a reduction in GSH, so that after 52 weeks of therapy, this biomarker was significantly lower in this group compared to the other groups.

## 4. Discussion

The aim of this study was to assess and compare the values of oxidative stress biomarkers and elements of the antioxidative defense system in psoriasis-suffering patients treated with different therapeutic approaches. The therapeutic option for one group was MTX, while the other two groups were treated with biological drugs, such as secukinumab, an IL-17A blocker, and ustekinumab, an IL-12/IL-23 blocker.

Figure 1 shows the gradual resolution of skin changes in patients treated with all applied therapies, as well as the pathohistological features of skin lesions before the introduction of therapy and after 52 weeks. Both followed disease activity scores, PASI and BSA, as well as patient-reported outcomes, DLQI and EARP, which were significantly reduced in all three groups of treated patients (Figure 2). These results are consistent with previous research [30,31,32]. Some previous studies showed that, although MTX and biologics reduced psoriasis severity scores, biological drugs were more effective [33,34].

All applied therapeutic options induced a reduction in O_2_^−^ values (Figure 3a). On the other hand, MTX induced a decrease in SOD activity, while both secukinumab and ustekinumab improved SOD activity (Figure 3b). The main recognized mechanism of the action of MTX is the inhibition of dihydrofolate reductase and further affection of DNA synthesis, resulting in a decrease in cell proliferation and the promotion of apoptosis [35]. Zimmerman and colleagues indicated the possibility of MTX to directly scavenge O_2_^−^ but not H_2_O_2_ [36]. Furthermore, MTX reduces inflammation through a reduction in pro-inflammatory cytokines, such as IL-17, IFN-γ, and IL-6 [37]. Free radical scavenging and a reduction in inflammation could be the key pharmacological mechanisms related to the beneficial effects of MTX in various inflammatory conditions. On the other hand, the main limitations of MTX use are its hepatotoxic and nephrotoxic effects [38]. It is shown that a higher dose of MTX induced a reduction in SOD and GSH levels in the liver of MTX-treated mice and rats [39,40]. A previous study dealing with MTX effects in psoriatic patients showed an insignificant decrease in SOD and CAT activity combined with an increase in plasma MDA after 12 weeks of treatment [41]. Results of this study showed that MTX reduced the values of TBARS after 52 weeks of treatment (Figure 5a). It appears that the beneficial clinical effects of MTX and its side effects, at least partially, could arise from the balance of its antioxidative/anti-inflammatory and pro-oxidative/pro-inflammatory actions. Both applied biological drugs, secukinumab and ustekinumab, reduced O_2_^−^ production and improved SOD activity (Figure 3a,b). There is very limited data regarding the effects of biologics on redox balance. In a murine model of psoriasis, the blockage of IL-17A resulted in a significant reduction in oxidative stress and inflammation [42]. It was also shown that levels of IL-17A and the severity of the disease correlate with peripheral oxidative stress and vascular dysfunction. Another study showed that the ustekinumab-induced blockage of IL-23 in an experimental model of cerebral ischemia resulted in an improvement of brain SOD and glutathione peroxidase (GPx) activity [43]. These effects were achieved through JAK2-STAT3 in the JAK/STAT pathway. A study with a similar design also showed that cerebral ischemia induces an increase in MDA and a reduction in GPx, GSH, CAT, and SOD in an experimental model of brain ischemia–reperfusion injury [44]. The application of secukinumab resulted in a decrease in MDA and an improvement of GPx, GSH, CAT, and SOD levels. In our study, only MTX induced a reduction in CAT activity, while neither secukinumab nor ustekinumab had induced any significant change in H_2_O_2_ or CAT activity, although CAT activity was significantly lower in all psoriatic groups compared to healthy controls (Figure 4a,b, Table 3). 

Some of the previous studies showed increased levels of NO and NO_2_^−^ levels in patients with psoriasis [45,46]. Another study indicated reduced NO bioavailability and NO-dependent vasodilation in psoriatic patients [47]. The suggested mechanism assumes IL-17 mediated a reduction in endothelial NO synthase (NOS) activity [48]. Similarly, NO content and NOS activity were significantly lower in cells treated with M5 cytokines (TNF-α, IL-17A, IL-22, IL-1, and oncostatin-M) compared to normal cells [49]. In our study, NO_2_^−^ levels, as a marker of NO production, were similar in psoriasis-suffering patients and healthy individuals and did not change significantly during the therapeutic procedures (Figure 5b, Table 3). Unchanged values of NO_2_^−^ could be the consequence of increased production of O_2_^−^ and the formation of peroxynitrite (ONOO^−^), one of the most reactive and deleterious reactive species [50]. 

One of the major limitations of this study is the number of patients. Such results should be tested by monitoring a larger number of patients who are treated in several different centers in order to reach a more comprehensive conclusion.

This research represents a solid initial basis for future research that would further shed light on the role of oxidative stress in the pathogenesis of psoriasis and the impact of different therapeutic options. Results of this study indicate a more pronounced improvement of redox status in patients treated with biologics compared to MTX. Due to reduced antioxidative capacity in patients treated with MTX, shown by us and others [51], it would be of interest to consider antioxidants as supportive therapy. The aim of the application of antioxidants, either as adjuvants or maintenance therapy, should be directed towards the increase in endogenous antioxidative capacity, especially in MTX-treated patients. Furthermore, due to the close connection between inflammation, oxidative stress, and psoriasis, a higher intake of dietary antioxidants could be one of the first steps in the improvement of the clinical presentation of psoriatic patients [51]. Another clinical relevance of these results is potential use in diagnosis given that we also indicated a lower ability of the antioxidant defense system. Other authors also showed lower values of CAT and SOD in psoriatic patients compared to the other forms of inflammatory skin diseases [51].

## 5. Conclusions

Results of this study showed that the most significant changes in the redox balance of patients with psoriasis were related to the values of O_2_^−^ and SOD activity. Estimated biological drugs, secukinumab and ustekinumab, had a more pronounced antioxidant effect, which was mainly reflected in the reduction in O_2_^−^ levels and increase in SOD activity. Such results confirm the strong relation between oxidative stress and inflammation. Bearing in mind that reactive species also act as signal molecules and that their production increases in conditions of inflammation (which further worsens inflammation), the importance of oxidative stress, both in understanding the pathogenesis and as a potential therapeutic target, is certainly of great importance. All applied therapeutic options, MTX and biologics, induced a reduction in PASI and BSA, disease activity outcomes, and patient-reported outcomes, expressed through DLQI and EARP.

## Figures and Tables

**Figure 1 biomedicines-12-00587-f001:**
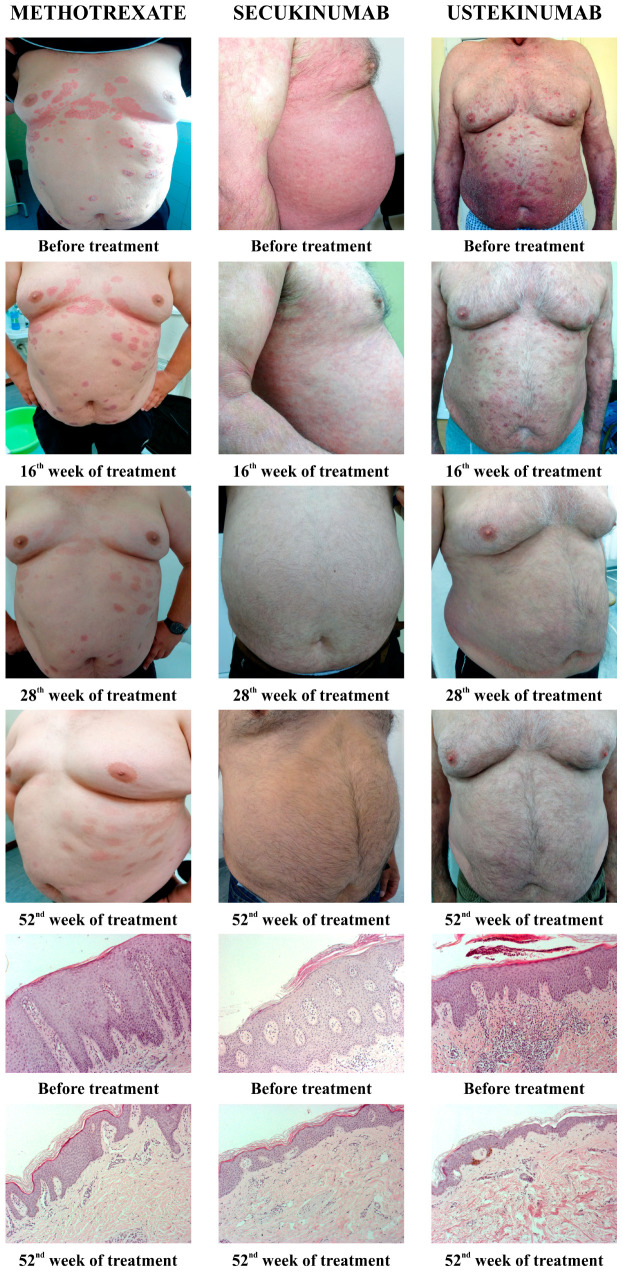
Presentation of clinical and histological changes in psoriatic lesions depending on the applied therapy. Clinical changes were shown before the introduction of therapy, as well as after 16, 28, and 52 weeks of therapy. The pathohistological appearance of psoriatic lesions was shown before the start of therapy and in the 52nd week.

**Figure 2 biomedicines-12-00587-f002:**
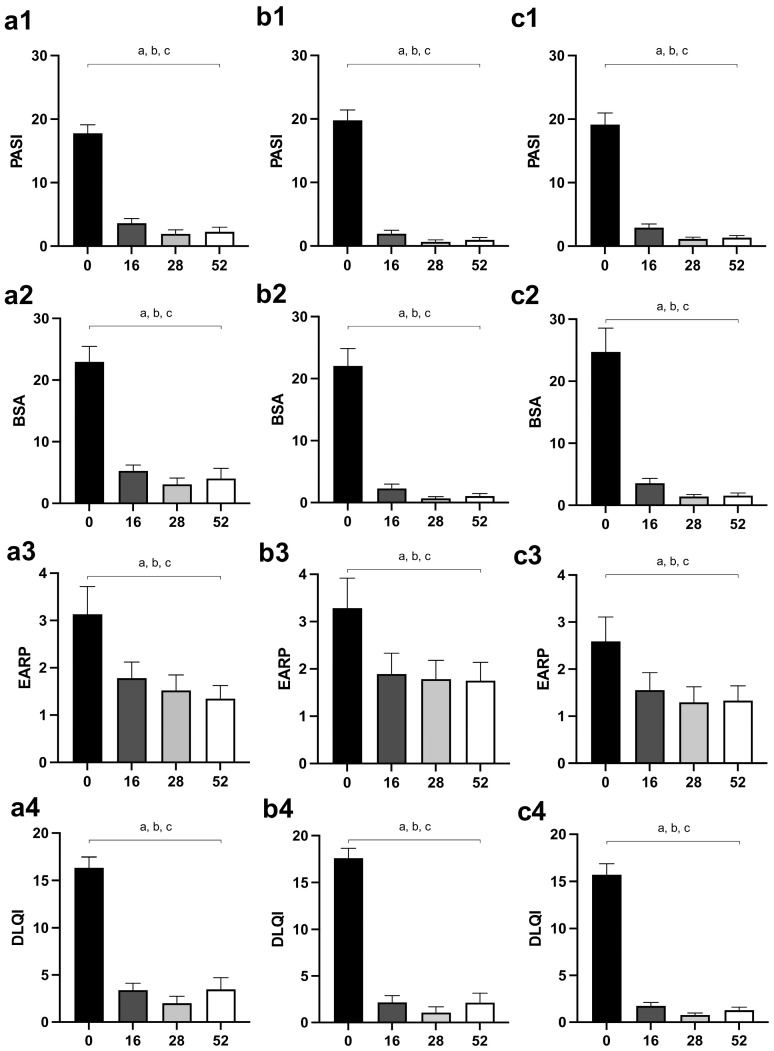
Changes of disease activity outcomes (PASI and BSA) and patient-reported outcomes (EARP and DLQI) in different groups of psoriatic patients—(**a1**–**a4**), patients treated with MTX; (**b1**–**b4**), patients treated with secukinumab; (**c1**–**c4**), patients treated with ustekinumab. The values are shown for points of interest: 0, before the introduction of therapy, as well as after 16, 28, and 52 weeks of therapy. A *p*-value of less than 0.05 (*p* < 0.05) was considered statistically significant. A, a—difference between the beginning of the therapy and the 16th week of the treatment; b—difference between the beginning of the therapy and the 28th week of the treatment; c—difference between the beginning of the therapy and the 52nd day of the treatment. Abbreviations: PASI—Psoriasis Area Severity Index; BSA—Body Surface Area; EARP—Early Arthritis for Psoriatic Patients; DLQI—Dermatology Life Quality Index; MTX—methotrexate.

**Figure 3 biomedicines-12-00587-f003:**
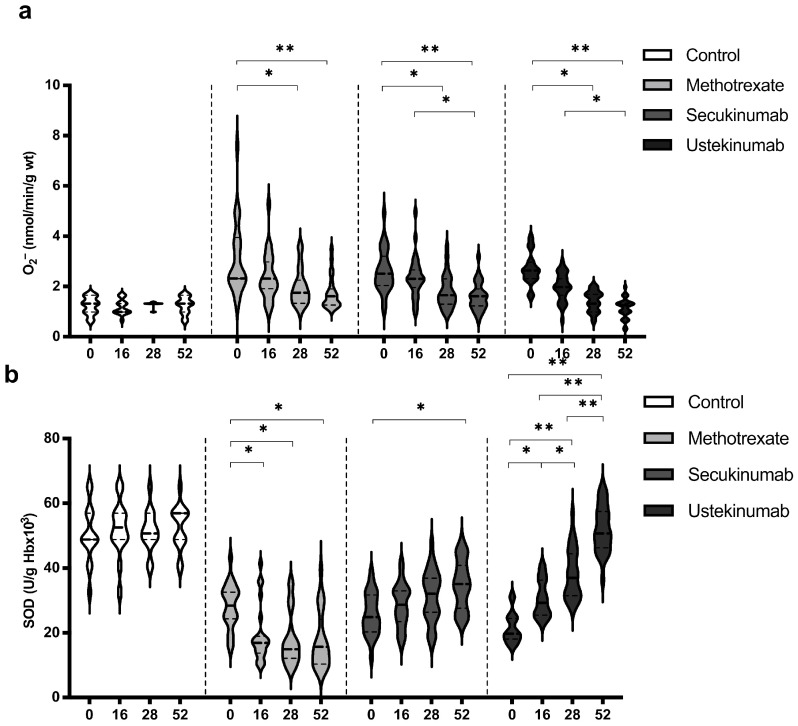
Values of (**a**) superoxide anion radical (O_2_^−^) and (**b**) activity of superoxide dismutase (SOD) in different groups of psoriatic patients. A *p*-value of less than 0.05 (*p* < 0.05) was considered statistically significant. Values are displayed in the form of violin plots to show the distribution of values within groups. *—*p* < 0.05; **—*p* < 0.01.

**Figure 4 biomedicines-12-00587-f004:**
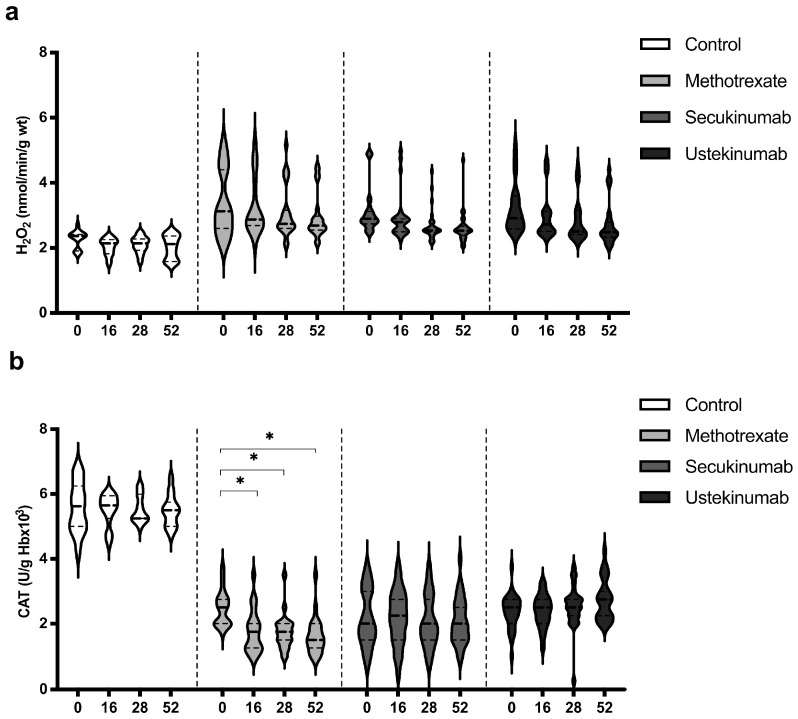
Values of (**a**) hydrogen peroxide (H_2_O_2_) and (**b**) activity of catalase (CAT) in different groups of psoriatic patients. A *p*-value of less than 0.05 (*p* < 0.05) was considered statistically significant. Values are displayed in the form of violin plots to show the distribution of values within groups. *—*p* < 0.05.

**Figure 5 biomedicines-12-00587-f005:**
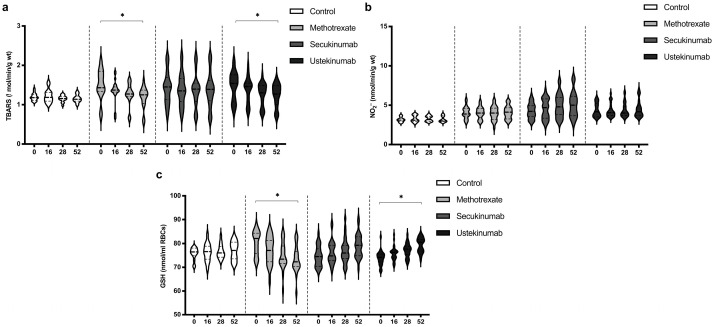
Values of (**a**) TBARS, (**b**) nitrites (NO_2_^−^), and (**c**) level of reduced glutathione (GSH) in different groups of psoriatic patients. A *p*-value of less than 0.05 (*p* < 0.05) was considered statistically significant. Values are displayed in the form of violin plots to show the distribution of values within groups. *—*p* < 0.05.

**Table 1 biomedicines-12-00587-t001:** Baseline characteristics of the study population.

	Healthy Controls	MTX Treated Patients	Secukinumab Treated Patients	Ustekinumab Treated Patients
Number of participants	15	23	28	27
Age	44.27 ± 11.61	46.70 ± 14.65	46.04 ± 13.95	46.00 ± 16.20
Female	5	7	8	9
Male	10	16	20	18
Female (%)	33.3	30.43	28.57	33.3
Male (%)	66.67	69.57	71.43	66.67

**Table 2 biomedicines-12-00587-t002:** Comparisons of O_2_^−^ and SOD values between the groups. A *p*-value of less than 0.05 was considered statistically significant (*—*p* < 0.05; **—*p* < 0.01).

Superoxide Anion Radical (O_2_^−^)		
	Before the initiation	52nd week
Control vs. MTX	*p* < 0.01 **	*p* > 0.05
Control vs. secukinumab	*p* < 0.01 **	*p* > 0.05
Control vs. ustekinumab	*p* < 0.01 **	*p* > 0.05
MTX vs. secukinumab	*p* > 0.05	*p* > 0.05
MTX vs. ustekinumab	*p* > 0.05	*p* > 0.05
Secukinumab vs. ustekinumab	*p* > 0.05	*p* > 0.05
**Superoxide Dismutase (SOD)**		
Control vs. MTX	*p* < 0.01 **	*p* < 0.01 **
Control vs. secukinumab	*p* < 0.01 **	*p* < 0.01 **
Control vs. ustekinumab	*p* < 0.01 **	*p* < 0.01 **
MTX vs. secukinumab	*p* > 0.05	*p* < 0.05 *
MTX vs. ustekinumab	*p* > 0.05	*p* < 0.01 **
Secukinumab vs. ustekinumab	*p* > 0.05	*p* < 0.01 **

Abbreviations: MTX—methotrexate.

**Table 3 biomedicines-12-00587-t003:** Comparison of H_2_O_2_ and CAT values between the groups. A *p*-value of less than 0.05 was considered statistically significant (*—*p* < 0.05; **—*p* < 0.01).

Hydrogen Peroxide (H_2_O_2_)		
	Before the initiation	52nd week
Control vs. MTX	*p* < 0.05 *	*p* < 0.05 *
Control vs. secukinumab	*p* < 0.05 *	*p* < 0.05 *
Control vs. ustekinumab	*p* < 0.05 *	*p* < 0.05 *
MTX vs. secukinumab	*p* > 0.05	*p* > 0.05
MTX vs. ustekinumab	*p* > 0.05	*p* > 0.05
Secukinumab vs. ustekinumab	*p* > 0.05	*p* > 0.05
**Catalase (CAT)**		
Control vs. MTX	*p* < 0.01 **	*p* < 0.01 **
Control vs. secukinumab	*p* < 0.01 **	*p* < 0.01 **
Control vs. ustekinumab	*p* < 0.01 **	*p* < 0.01 **
MTX vs. secukinumab	*p* > 0.05	*p* < 0.05 *
MTX vs. ustekinumab	*p* > 0.05	*p* < 0.05 *
Secukinumab vs. ustekinumab	*p* > 0.05	*p* < 0.05 *

Abbreviations: MTX—methotrexate.

**Table 4 biomedicines-12-00587-t004:** Comparison of TBARS, NO_2_^−^, and GSH values between the groups. A *p*-value of less than 0.05 was considered statistically significant (*—*p* < 0.05).

TBARS		
	Before the initiation	52nd week
Control vs. MTX	*p* > 0.05	*p* > 0.05
Control vs. secukinumab	*p* > 0.05	*p* > 0.05
Control vs. ustekinumab	*p* > 0.05	*p* > 0.05
MTX vs. secukinumab	*p* > 0.05	*p* > 0.05
MTX vs. ustekinumab	*p* > 0.05	*p* > 0.05
Secukinumab vs. ustekinumab	*p* > 0.05	*p* > 0.05
**Nitrites (NO_2_^−^)**		
Control vs. MTX	*p* > 0.05	*p* > 0.05
Control vs. secukinumab	*p* > 0.05	*p* > 0.05
Control vs. ustekinumab	*p* > 0.05	*p* > 0.05
MTX vs. secukinumab	*p* > 0.05	*p* > 0.05
MTX vs. ustekinumab	*p* > 0.05	*p* > 0.05
Secukinumab vs. ustekinumab	*p* > 0.05	*p* > 0.05
**Reduced glutathione (GSH)**		
Control vs. MTX	*p* < 0.05 *	*p* < 0.05 *
Control vs. secukinumab	*p* > 0.05	*p* > 0.05
Control vs. ustekinumab	*p* > 0.05	*p* > 0.05
MTX vs. secukinumab	*p* < 0.05 *	*p* < 0.05 *
MTX vs. ustekinumab	*p* < 0.05 *	*p* < 0.05 *
Secukinumab vs. ustekinumab	*p* > 0.05	*p* > 0.05

Abbreviations: MTX—methotrexate.

## Data Availability

The authors declare that there are no conflicts of interest regarding the publication of this article.

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
