# Peer review of "Effects of Different Therapeutic Approaches on Redox Balance in Psoriatic Patients"

_biomedicines, 2024, doi:10.3390/biomedicines12030587_

Round 1

Reviewer 1 Report

Comments and Suggestions for Authors

The Authors aimed to assess the effects of different therapeutic approaches, methotrexate, secukinumab and ustekinumab on systemic oxidative stress biomarkers in psoriatic patients. The article is written clearly. The results justify the conclusions.

Some minor remarks are as follows:

-        The following statement in the Abstract section needs to be written more clearly: “Biologics improved GSH level, ustekinumab significantly, in 40 the 52nd week, while methotrexate induced significant reduction of GSH”.

-        The Authors stated in the Introduction section that oxidative stress is recognized as step in the pathophysiological cascade of psoriasis and emphasized the close interconnection between inflammation and oxidative stress in psoriasis. In line with this, the Authors are referred to a recent publications: https://doi.org/10.5937/jomb0-45076 (i.e., higher AOPP and CAT actiavity as compared with healthy controls).

-        The Discussion section could be extended with more information on modifiable risk factors in psoriasis. In line with this, the Authors are referred to a recently published article related to the antioxidants in skin diseases: https://doi.org/10.3390/antiox12101875.  The Authors of the latter article suggested that age and gender-specific therapeutic antioxidant approaches (specifically in men) could be helpful in patients with psoriasis and atopic dermatitis and suggested the importance of dietary habits containing food rich in antioxidants, as well as the use of antioxidants as a potential therapeutic strategy in the treatment of the examined skin inflammatory diseases.

-        The abbreviations need to be explained when first mentioned in the text.

Comments on the Quality of English Language

-        Minor editing of English language is recommended.

Author Response

- The following statement in the Abstract section needs to be written more clearly: “Biologics improved GSH level, ustekinumab significantly, in 40 the 52nd week, while methotrexate induced significant reduction of GSH”.

Thank you for your suggestion. We have clarified the sentence you marked so that it is now clearer: "Ustekinumab induced significant increase in GSH level after 52 weeks of treatment, while methotrexate reduced GSH.". Please see the abstract, the lines 40-41.

- The Authors stated in the Introduction section that oxidative stress is recognized as step in the pathophysiological cascade of psoriasis and emphasized the close interconnection between inflammation and oxidative stress in psoriasis. In line with this, the Authors are referred to a recent publications: https://doi.org/10.5937/jomb0-45076 (i.e., higher AOPP and CAT actiavity as compared with healthy controls).

Thank you. We cited the publication you pointed to, given that it deals with oxidative stress in psoriasis suffering patients. Please see the Introduction section, line 91 and reference list.

- The Discussion section could be extended with more information on modifiable risk factors in psoriasis. In line with this, the Authors are referred to a recently published article related to the antioxidants in skin diseases: https://doi.org/10.3390/antiox12101875.  The Authors of the latter article suggested that age and gender-specific therapeutic antioxidant approaches (specifically in men) could be helpful in patients with psoriasis and atopic dermatitis and suggested the importance of dietary habits containing food rich in antioxidants, as well as the use of antioxidants as a potential therapeutic strategy in the treatment of the examined skin inflammatory diseases.

Discussion was extended due to your suggestion. Please see discussion section, the lines 485-496 and reference list.

- The abbreviations need to be explained when first mentioned in the text.

The abbreviations were checked and additionally explained in the text.

Reviewer 2 Report

Comments and Suggestions for Authors

Dear authors,

this manuscript clearly describes the aims of your study and the structure of it is complete. I suggest you some advice to improve your manuscript:

- change the photo of Figure 1 of the patient of the MTX group because it shows legs while the photos of the other two groups show psoriasis involving the trunk

- improve the description of Figure 2 explaining the names of the three groups  

- make more emphasis in the discussion on the relevance of the variation of the oxidative markers and explain why this could be useful for therapeutic or diagnostic approaches.

Comments on the Quality of English Language

The quality of English is good and this manuscript needs only some minor revisions to the vocabulary used.

Author Response

- change the photo of Figure 1 of the patient of the MTX group because it shows legs while the photos of the other two groups show psoriasis involving the trunk.

Thank you for your suggestion. The photos regarding the MTX group in Figure 1 were changed. Please see Figure 1.

- improve the description of Figure 2 explaining the names of the three groups.

The description of Figure 2 is improved. Please see the results section and caption for Figure 2, lines 329-334.

- make more emphasis in the discussion on the relevance of the variation of the oxidative markers and explain why this could be useful for therapeutic or diagnostic approaches.

We further clarified the potential clinical significance of the obtained results. Please see discussion section lines 485-496, and reference list.
